# Depression in Male Inmates

**Dimitrios Kastos ***, **Evangelos Dousis** , **Afroditi Zartaloudi** , **Niki Pavlatou** , **Antonia Kalogianni,**
**Georgia Toulia, Vasiliki Tsoulou and Maria Polikandrioti ***

Department of Nursing, University of West Attica, 12243 Athens, Greece
* Correspondence: acn19005@uniwa.gr (D.K.); mpolik2006@yahoo.com (M.P.)

**Abstract:** Introduction: The prevalence of psychiatric morbidity is high among incarcerated individuals. Severe mental disorder is five to ten times higher among prisoners compared to the general population. Several factors are held to be responsible for the high prevalence of depression in prison: mainly poor living conditions (narrow room, loss of privacy), limited interpersonal relationships, and lack of mental health access. Inmates are at increased risk of all-cause mortality, suicide, self-harm, violence, and victimization while those with mental disorders are involved in conflicts and are more likely to be charged with prison rules. Purpose: To explore depression among male inmates. Methods and material: In the study, 101 male inmates were enrolled. Data were collected by the completion of a "self-rating depression scale (SDS)-Zung" which included participants' characteristics. The statistical significance level was $p < 0.05$. Results: Of the 101 participants, 51.4% of inmates were under 40 years old, 54.5% were married, 45.6% had been convicted of homicide and 38.6% had a life sentence. Normal depression levels were experienced by 62.4% of the participants, while 21.8% were mildly depressed, 14.9% were moderately depressed and 1.0% severely depressed. Foreign prisoners had statistically significant higher scores of depression compared to Greeks (median 48 vs. 45, $p = 0.012$); those suffering from a chronic disease compared to those who did not (median 48 vs. 45, $p = 0.038$); those who had spent time in solitary confinement compared to those who had not (median 46 vs. 43.5, $p = 0.038$) as well as those who had not considered harming themselves compared to those who had thought of it (median 46 vs. 44, $p = 0.017$). Conclusion: Given that prison populations are marginalized and deprived of the rights that people in the community benefit from, establishing the prevalence of depression in male inmates and its associated characteristics may help to formulate recommendations for future prison health care services. Clinical, research, and policy efforts are needed to improve prison mental health.

**Keywords:** depression; Zung; inmates

## 1. Introduction

During recent years, the number of people incarcerated has reached more than 10.35 million worldwide. Since 2000, this number has been increased by 30%, a rate higher than the general population, which globally increased by 20% in the same period. In greater detail, from 2013 the prison population has increased in the USA, Asia, Europe and Oceania [1].The USA has reported one in five of the approximately 11 million incarcerated in 2019 globally [2].

Inmates are at greater risk of mental illness due to several factors such as loss of personal freedom and privacy, prison violence, social isolation, lack of or diminished access to mental health care, substance abuse, chronic health problems, family history of mental illness, overcrowding, and memorizing illegal acts that make them feel guilty [1,3,4]. Mental illness may be present at the time of admission to prison or develop during imprisonment and deteriorate due to the prison environment and conditions of detention [5].

According to worldwide estimates, one in nine prisoners experiences a mental disorder, mainly depression [1]. The prevalence of depression among prisoners was 42%, 43%, 81%,

85%, 35.3% and 37% in Iran, New York, India, Pakistan, Nepal, and Nigeria, respectively [4] as well as 19.2% in Norway and 57.4% in Turkey, respectively [3].

Men with depression experience behavioral problems which create difficulties in adaptation to prison and may in turn deteriorate their mental health state. For example, men tend to be more irritable and aggressive, engage in conflicts or risky activities, and have alcohol use disorder [6]. Mental disorder is three times higher in male prisoners than in the general Greek adult population [7]. Greek male inmates are more likely to report getting into a fight [5].

One of the important reasons to screen depression is the danger of suicide [8–10]. Self-harm is a leading cause of morbidity in prisoners with an annual presence of 5–6% in men. Prisoners who self-harm are at a six-to-eight-times increased risk of suicide while incarcerated and this persists after release [8]. Among Greek male prisoners, psychiatric condition, illicit substance use and aggression were significant risk factors in self-injured behavior [9].

Understanding the profile of male inmates with depression contributes to the identification of demands in treatment and rehabilitation. Thus, the aim of this cross-sectional study was to explore depression among male inmates and the associated self-reported characteristics.

## 2. Methods and Material

### 2.1. Design, Setting, and Period of the Study

In the study, 101 male inmates were enrolled in the prefecture of Attica during the period January–February 2021. This was a cross-sectional descriptive study. Participants were selected by the method of convenience sampling.

### 2.2. Sample: Inclusion and Exclusion Criteria

Criteria for inclusion of participants in the study were: (a) to understand the Greek language and (b) to have a translator available in the case of a non-Greek participant. Exclusion criteria were inmates: (a) not cooperating in data completion, (b) being kept in solitary confinement for the past six months and (c) being transferred recently from another prison unit.

### 2.3. Data Collection and Procedure

The collection of data was performed by the method of interview using a research instrument designed to serve the purposes of the study. To prevent potential interference, inmates were invited to a private office room and assessed separately. Completion of each questionnaire was carried out in the afternoon, when participants had no other activity. The duration of each interview was approximately 45 min.

### 2.4. Research Instrument

The instrument used was a questionnaire, which included self-reported characteristics and the self-rating depression scale Zung (SDS).

In terms of demographic characteristics, the following data were collected: age, marital status, education level, job, residency, nationality, the existence of children or siblings and whether they had served in the military.

Regarding characteristics related to prison, the following data were recorded: reason for imprisonment, length of sentence, visits by family and friends, ways of spending free time, work at prison, solitary confinement, desire to be isolated from other prisoners, satisfaction with health state in prison and major stress.

The following clinical characteristics were reported: chronic and communicable disease, psychiatric history, history of substance abuse, alcohol consumption before prison and current smoking. With reference to suicidality, it was recorded if they had attempted suicide and if they had ever self-harmed themselves.

As far as characteristics of abuse before imprisonment is concerned, the following were recorded: physical, psychological and sexual abuse and memory of anxiety-, anger-, or sadness-inducing events.

### 2.5. Assessment of Depression

The "self-rating depression scale (SDS)-Zung" was used to assess depression in psychiatric male inmates. The SDS scale consists of 20 questions that evaluate how respondents felt during the previous week. Respondents have the ability to answer each question on a four-point Likert-type scale. In five questions it is first necessary to reverse the scores. The scores attributed to the questions are summed up leading to a final score ranging from 20 to 80. Higher scores indicate higher levels of depression [11,12].

### 2.6. Ethical Considerations

The study was approved by the Ethical Committee where it was conducted. Inmates who met the entry criteria were informed by the researcher of the purposes of this study. All inmates participated only after they had given their written consent. Data collection guaranteed anonymity and confidentiality. All subjects were informed of their rights to refuse or discontinue participation in the study, according to the ethical standards of the Declaration of Helsinki (1989) of the World Medical Association.

### 2.7. Statistical Analysis

Categorical data are presented with absolute and relative (%) frequencies, while continuous ones are presented with mean, standard deviation, and a median and interquartile range (IQR). Kruskal–Wallis and Mann–Whitney tests were performed to evaluate the association between depression and the inmate's characteristics. Normality was tested with Kolmogorov–Smirnov και graphically with histograms. Reliability was assessed with a Cronbach's a estimate. In addition, multiple linear regression was performed to assess the impact of an inmate's characteristics (independent factors) on their depression. Results are presented as β coefficients and a 95% confidence interval (95% CI). The observed significance level of 5% was considered statistically significant. All statistical analyses were performed with version 25 of the SPSS statistical program (SPSSInc, Chicago, IL, USA).

## 3. Results

### 3.1. Sample Description

Table 1 shows the demographic characteristics of the sample. A total of 51.4% of inmates were under 40 years old, 54.4% were married, 57.4% had a secondary education, 19.8% were unemployed, 43.6% lived in Attica, 84.1% were Greeks, 60.4% had no children, 85.1% had siblings and 65.3% had served in the military.

**Table 1.** Demographic characteristics of inmates (*n* = 101).

|  | *n* (%) |
|---|---|
| Age (years) | |
| 17–20 | 6(5.9%) |
| 21–30 | 16(15.8%) |
| 31–40 | 30(29.7%) |
| 41–50 | 29(28.7%) |
| 51–60 | 16(15.8%) |
| >60 | 4(4.0%) |
| Marital Status | |
| Single | 20(19.8%) |
| Married | 55(54.4%) |
| Divorced | 23(22.8%) |
| Widowed | 2(2.0%) |
| Living Together | 1(1.0%) |

**Table 1.** *Cont.*

| | |
|---|---|
| Education Level | |
| Primary School | 22(21.8%) |
| High School | 58(57.4%) |
| University | 15(14.9%) |
| No education | 6(5.9%) |
| Job | |
| Unemployed | 20(19.8%) |
| Civil Servant | 5(5.0%) |
| Private Employee | 29(28.6%) |
| Freelancer | 42(41.6%) |
| Pensioner | 5(5.0%) |
| Residency | |
| Attica | 44(43.6%) |
| County capital | 20(19.8%) |
| Small Town | 16(15.8%) |
| Village | 21(20.8%) |
| Nationality | |
| Greek | 85(84.1%) |
| Albanian | 8(7.9%) |
| Bulgarian | 3(3.0%) |
| Other | 5(5.0%) |
| No of children | |
| 0 | 61(60.4%) |
| 1 | 11(10.9%) |
| 2 | 21(20.8%) |
| >2 | 8(7.9%) |
| Siblings (yes) | 86(85.1%) |
| Served in military (yes) | 66(65.3%) |

In Table 2, the data presented concern the characteristics related to prison. The main reason for imprisonment was homicide (45.6%); 38.6% had been sentenced to life, 62.4% were able to visit their environment, 50.5% spent their free time watching television, 39.6% worked during prison hours, 24.8% were very satisfied with their health state, 74.3% spent time in solitary confinement while 40.6% wanted isolation from other inmates and 50.5% acknowledged the future after leaving prison as their major cause of stress. Of the 101 participants, 38.6% had attempted suicide, while 41.6% had harmed themselves.

**Table 2.** Distribution of the sample according to characteristics related to imprisonment (*n* = 101).

| | *n* (%) |
|---|---|
| Reason for imprisonment | |
| Financial | 2(2.0%) |
| Theft–robbery | 17(16.8%) |
| Drug trafficking | 7(6.9%) |
| Attempted murder | 11(10.9%) |
| Injuries | 10(9.9%) |
| Homicide | 46(45.6%) |
| Rape | 7(6.9%) |
| Migrant Trafficking | 1(1.0%) |
| Sentence (years) | |
| Under trial | 21(20.8%) |
| ≤10 | 24(23.8%) |
| >10 | 17(16.8%) |
| Life Sentence | 39(38.6%) |
| Visited by someone (yes) | 63(62.4%) |
| Who visits you? | |
| Parents | 31(49.2%) |
| Friends | 8(12.7%) |

**Table 2.** *Cont.*

| | |
|---|---|
| Relatives | 14(22.2%) |
| Children | 3(4.8%) |
| Partner | 7(11.1%) |
| How do you spend your free time? | |
| Reading | 15(14.8%) |
| TV | 51(50.5%) |
| Painting | 1(1.0%) |
| Music | 25(24.8%) |
| Physical exercise | 9(8.9%) |
| Do you work during prison time (yes) | 40(39.6%) |
| Satisfied with your health state? | |
| Very | 25(24.8%) |
| Enough | 42(41.6%) |
| A little | 27(26.7%) |
| Not at all | 7(6.9%) |
| Have you been in solitary confinement? | |
| Yes | 75(74.3%) |
| No | 26(25.7%) |
| Do you desire isolation from other prisoners? | |
| Yes | 41(40.6%) |
| No | 53(52.5%) |
| Sometimes | 7(6.9%) |
| What is your major cause of stress? | |
| My health | 21(20.8%) |
| Relationships with other prisoners | 11(10.9%) |
| Relationships with family | 18(17.8%) |
| Future after prison | 51(50.5%) |

In Table 3, the data presented concern clinical characteristics. Specifically, 14.9% suffered from a chronic disease and 11.9% from a communicable disease, while 74.3% had a history of psychiatric problems, 58.4% had a history of substance abuse and 35.6% and 85.1% were consuming alcohol before prison and were current smokers, respectively. Moreover, 38.6% had attempted suicide and 41.6% had harmed themselves.

**Table 3.** Distribution of the sample according to clinical characteristics (*n* = 101).

| | *n* (%) |
|---|---|
| Do you suffer from a chronic disease (yes) | 15(14.9%) |
| Do you have a communicable disease (yes) | 12(11.9%) |
| What communicable disease do you have? | |
| HIV | 3(25.0%) |
| Hepatitis | 9(75.0%) |
| Do you have a psychiatric history (yes) | 75(74.3%) |
| Do you have a history of substance abuse (yes) | 59(58.4%) |
| Smoker (yes) | 86(85.1%) |
| Consuming alcohol before prison | |
| Yes | 36(35.6%) |
| No | 24(23.8%) |
| Sometimes | 41(40.6%) |
| Have you attempted suicide? | |
| Yes | 39(38.6%) |
| No | 57(56.4%) |
| Don't want to answer | 5(5.0%) |
| Have you ever harmed yourself? | |
| Yes | 42(41.6%) |
| No | 59(58.4%) |

In Table 4 are presented data concerning any type of abuse before prison. Of the 101 participants, 54.5%, 12.9%, and 59% had been physically, sexually, and psychologically abused, respectively.

**Table 4.** Distribution of the sample according to abuse before imprisonment (*n* = 101).

|  | *n* (%) |
|---|---|
| Have you been physically abused? (yes) | 55(54.5%) |
| Who physically abused you? |  |
| Father | 27(49.1%) |
| Mother | 10(18.1%) |
| Relative | 6(10.9%) |
| Friend | 3(5.5%) |
| Unknown | 9(16.4%) |
| How many times were you physically abused? |  |
| Once | 12(21.8%) |
| Twice | 3(5.5%) |
| More than twice | 40(72.7%) |
| Have you mentioned the physical abuse to someone? (yes) | 12(21.8%) |
| Have you been sexually abused? (yes) | 13(12.9%) |
| Who sexually abused you? |  |
| Father | 1(7.7%) |
| Mother | 1(7.7%) |
| Relative | 1(7.7%) |
| Friend | 6(46.2%) |
| Unknown | 4(30.7%) |
| How many times were you sexually abused? |  |
| Once | 7(53.8%) |
| Twice | 1(7.7%) |
| More than twice | 5(38.5%) |
| Have you mentioned the sexual abuse to someone? (yes) | 4(30.8%) |
| Have you been psychologically abused? (yes) | 59(59.0%) |
| Who abused you psychologically? |  |
| Father | 13(22.0%) |
| Mother | 12(20.3%) |
| Grandfather/Grandmother | 1(1.7%) |
| Relative | 12(20.4%) |
| Friend | 13(22.0%) |
| Unknown | 8(13.6%) |
| How many times were you psychologically abused? |  |
| Once | 8(13.6%) |
| Twice | 3(5.1%) |
| More than twice | 48(81.4%) |
| Have you mentioned the psychological abuse to someone? (yes) | 15(25.4%) |

From Table 5, concerning depression, we observe that at least 50% of inmates scored below 45 (median) on the Zung depression scale. In addition, 25% of inmates scored below 42. The average score on the Zung scale was 45.1 $\pm$ 4.1. Reliability of participantss answers was estimated as high (Cronbach's a: 0.840).

**Table 5.** Measuring the depression of inmates (*n* = 101).

|  | Mean (SD) | Median (IQR) | Cronbach's a |
|---|---|---|---|
| Depression (Range 20–80) | 45.1(4.1) | 45(42–48) | 0.840 |

| Depression Levels | *n* (%) |  |  |
|---|---|---|---|
| Normal (score < 50) | 63(62.4%) |  |  |
| Mildly Depressed (score 50–59) | 22(21.8%) |  |  |
| Moderately Depressed (score 60–69) | 15(14.9%) |  |  |
| Severely Depressed (score $\geq$ 70) | 1(1.0%) |  |  |

SD: Standard Deviation, IQR: Interquartile Range.

Normal depression levels were experienced by 62.4% of the participants, while 21.8% were mildly depressed, 14.9% moderately depressed, and 1.0% severely depressed.

### 3.2. Factors Associated with Depression

Table 6 shows the association between depression and inmates' characteristics.

**Table 6.** Factors associated with inmates' depression.

|  | Mean (SD) | Median (IQR) | *p*-Value * |
|---|---|---|---|
| Age |  |  | 0.537 |
| <40 | 45.5(3.6) | 45(42.5–48) |  |
| 41–50 | 44.7(5.5) | 44(41–48) |  |
| >50 | 44.6(3.3) | 44.5(42.5–46.5) |  |
| Marital status |  |  | 0.759 |
| Single | 45.6(4.1) | 46(43–48) |  |
| Married | 45.1(3.9) | 45(42–48) |  |
| Divorced/Widowed | 44.7(4.7) | 46(41–48) |  |
| Education Level |  |  | 0.221 |
| No education/Primary School | 46.1(3.3) | 46(44–48) |  |
| High School | 44.7(4.6) | 45(41–48) |  |
| University | 44.8(3.6) | 44(43–48) |  |
| Job |  |  | 0.392 |
| Unemployed | 46.0(4.4) | 46(42.5–48) |  |
| Employee | 44.8(4.1) | 45(42–48) |  |
| Residency |  |  | 0.541 |
| Attica | 45.7(4.4) | 45.5(42–49) |  |
| County capital | 44.4(4.0) | 44.5(41.5–46.5) |  |
| Small Town/Village | 44.8(3.9) | 45(43–48) |  |
| Nationality |  |  | 0.012 |
| Greek | 44.6(3.8) | 45(42–47) |  |
| Other | 47.6(4.9) | 48(45–49.5) |  |
| Children |  |  | 0.360 |
| No | 44.9(4.0) | 45(42–47) |  |
| Yes | 45.4(4.4) | 46(42.5–48) |  |
| Siblings |  |  | 0.333 |
| Yes | 45.0(4.3) | 45(42–48) |  |
| No | 46.0(3.4) | 47(42–48) |  |
| Served in military |  |  | 0.113 |
| Yes | 44.6(4.1) | 44(42–48) |  |
| No | 46.1(4.2) | 46(43–49) |  |
| Reason for imprisonment |  |  | 0.660 |
| Financial/Theft/Robbery | 45.6(3.9) | 46(42–48) |  |
| Drug trafficking/Migrant trafficking | 46.0(5.0) | 48(41.5–49.5) |  |
| Attempted murder/Homicide | 44.9(3.8) | 45(42–47) |  |
| Injuries/Rape | 44.8(5.2) | 44(42–48) |  |
| Sentence (years) |  |  | 0.686 |
| Under trial | 45.8(4.8) | 45(43–48) |  |
| ≤10 | 44.6(3.5) | 45(41.5–47.5) |  |
| >10 | 44.3(3.6) | 45(41–48) |  |
| Life Sentence | 45.4(4.4) | 46(42–49) |  |
| Visited by someone |  |  | 0.520 |
| Yes | 45.3(3.9) | 46(42–48) |  |
| No | 44.8(4.6) | 45(43–47) |  |
| How do you spend your free time? |  |  | 0.737 |
| Reading | 45.7(3.5) | 46(43–48) |  |
| TV | 44.8(3.9) | 45(42–48) |  |
| Music | 45.1(4.4) | 45(42–46) |  |
| Physical exercise | 45.6(6.0) | 49(43–49) |  |
| What is your major cause of stress? |  |  | 0.583 |
| My health | 45.7(4.9) | 44(43–47) |  |
| Relationships with other prisoners | 46.1(2.8) | 46(43–48) |  |

**Table 6.** *Cont.*

| | Mean (SD) | Median (IQR) | *p*-Value * |
|---|---|---|---|
| Relationships with family | 45.4(4.8) | 47(42–49) | |
| Future after prison | 44.6(3.8) | 45(42–48) | |
| Are you satisfied with your health state? | | | 0.764 |
| Very | 45.6(5.1) | 46(41–50) | |
| Enough | 44.9(3.2) | 45(43–48) | |
| A little/Not at all | 45.0(4.5) | 45(42–48) | |
| Have you been in solitary confinement? | | | 0.038 |
| Yes | 45.6(4.3) | 46(43–48) | |
| No | 43.7(3.5) | 43.5(41–46) | |
| Do you desire isolation from other prisoners? | | | 0.301 |
| Yes/Sometimes | 45.7(4.5) | 45.5(42–48.5) | |
| No | 44.5(3.8) | 45(42–47) | |
| Do you work during prison? | | | 0.558 |
| Yes | 44.7(4.1) | 44.5(42–48) | |
| No | 45.4(4.2) | 45(43–48) | |
| Do you suffer from a chronic disease? | | | 0.038 |
| Yes | 46.9(3.0) | 48(44–48) | |
| No | 44.8(4.2) | 45(42–48) | |
| Do you have a communicable disease? | | | 0.992 |
| Yes | 44.8(3.5) | 46(43–47.5) | |
| No | 45.1(4.2) | 45(42–48) | |
| Do you have a psychiatric history? | | | 0.211 |
| Yes | 44.9(4.2) | 45(42–48) | |
| No | 45.8(4.0) | 46(44–49) | |
| Do you have a history of substance abuse? | | | 0.838 |
| Yes | 45.1(3.7) | 45(42–48) | |
| No | 45.1(4.8) | 45(42–48) | |
| Current Smoker | | | 0.625 |
| Yes | 45.0(4.3) | 45(42–48) | |
| No | 45.6(2.8) | 46(44–46) | |
| Consuming alcohol before prison | | | 0.435 |
| Yes/Sometimes | 45.2(4.1) | 45(42–48) | |
| No | 44.7(4.4) | 45(41.5–47) | |
| Remember an event of anxiety/anger/sadness | | | 0.111 |
| Yes | 45.5(4.3) | 45(42–48) | |
| No | 43.9(3.4) | 43.5(41–46.5) | |
| Have you been physically abused? | | | 0.529 |
| Yes | 45.3(4.3) | 45(42–48) | |
| No | 44.9(4.0) | 44.5(42–48) | |
| Have you been sexually abused? | | | 0.525 |
| Yes | 45.8(3.3) | 45(44–48) | |
| No | 45.0(4.3) | 45(42–48) | |
| Have you been psychologically abused? | | | 0.204 |
| Yes | 44.8(4.3) | 44(42–48) | |
| No | 45.7(4.0) | 46(43–48) | |
| Have you attempted suicide? | | | 0.857 |
| Yes | 45.1(5.0) | 45(41–48) | |
| No | 45.2(3.7) | 46(43–48) | |
| Have you ever harmed yourself? | | | 0.342 |
| Yes | 45.6(4.8) | 46(42–49) | |
| No | 44.7(3.6) | 44.5(42–48) | |

* *p*-values were estimated using Mann–Whitney and Kruskal–Wallis tests.

A statistically significant correlation was observed between the inmates' depression score and their nationality (*p* = 0.012), whether they suffered from a chronic disease (*p* = 0.038) and whether they spent time in solitary confinement (*p* = 0.038). More specifically, foreign inmates experienced higher levels of depression (median 48) than Greeks

(median 45). Inmates suffering from a chronic illness experienced higher levels of depression (median 48) than those who did not (median 45). Inmates who spent time in solitary confinement experienced higher levels of depression (median 46) than those who did not (median 43.5).

### 3.3. Impact of Inmates' Characteristics on Depression

Multiple linear regression was performed to assess the impact of inmate characteristics (independent factors) on the depression they experienced (dependent variable).

In Table 7, we observed that foreign prisoners had a 2.8-point higher depression score than Greeks (β = 2.77, 95% CI: 0.65–4.88, *p* = 0.011). In addition, inmates with a chronic disease had a 2.9-point higher depression score than those without (β = 2.85, 95% CI: 0.61–5.09, *p* = 0.013).

**Table 7.** Impact of inmates' characteristics on their depression.

|  | β Coefficient (95% CI) | *p*-Value |
|---|---|---|
| Nationality |  |  |
| Greek | Ref.Cat. |  |
| Other | 2.77(0.65–4.88) | 0.011 |
| Do you suffer from a chronic disease? |  |  |
| No | Ref.Cat. |  |
| Yes | 2.85(0.61–5.09) | 0.013 |
| Have you been in solitary confinement? |  |  |
| No | Ref.Cat. |  |
| Yes | 1.23($-$0.55–3.00) | 0.174 |

## 4. Discussion

The present results illustrated that 21.8% and 14.9% were mildly and moderately depressed. In Brazil, Andreoli et al. [13] showed that among 1192 male prisoners the lifetime and 12-month prevalence rates for severe mental disorders (psychotic, bipolar disorders, severe depression) were 12.3% and 6.3%, respectively. In Ethiopia, (Amhara Region), Reta et al. [14], in a sample of 336 prisoners (98% men), showed that 44% experienced depression. Similarly, in Ethiopia (Bahir Dar city), depression accounted for 45.5% among 402 prisoners, of whom the majority were men (*n* = 394) [1]. In India, among 146 male inmates, 25.7% had moderately severe depression and 27.6% severe depression [15]. In China (Guangdong), depression rates were 28.8% among 1484 male prisoners [16]. A relevant study in Greece of 495 male prisoners indicated that 223 (45.06%) were diagnosed with a psychiatric disorder with the most prevalent reason for imprisonment being non-violent crimes (40.7%) [7].

All aforementioned results demonstrate that depression in prison varies globally, which is attributed to socioeconomic and cultural discrepancies between countries, to medical and judicial systems, to levels of prison infrastructure and health care delivery as well as to differences in the methodology of research studies (screening tools for assessing depression, sample size and study period) [3,10].

Psychiatric morbidity rates underscore the need for delivery of effective mental health care to the incarcerated population. Although depression is a treatable condition, in some countries it is doubtful if the majority of mentally ill inmates receive any treatment [10].Additionally, the present results showed that foreign prisoners, those suffering from a chronic illness, or those kept in solitary confinement were more depressed. Relevant studies have shown several other factors related with depression. For example, there were higher levels of depression among inmates who were divorced, had graduated from higher education, experienced major stressful events, were sentenced to five to 10 years, or those with a history of suicide attempts or chronic illness [14].

In addition, higher education, being accused of a crime, and low social support were predictors of depression [15]. Furthermore, a history of drinking problems and gambling addiction were related to depressive symptoms [16].

Though chronic medical illnesses, such as hypertension, epilepsy, asthma, cancer, tuberculosis, hepatitis C and human immunodeficiency virus (HIV), can be managed they cannot however be cured and this puts prisoners at a disadvantage. Additionally, more, inmates with chronic diseases are in need of a diet tailored to the illness's specific needs but in some countries they may receive only one meal a day [17]. Moreover, prisoners may suffer from chronic diseases due to adverse living conditions. For example, the density of the prison population is responsible for certain diseases, mainly infectious ones [17]. Meanwhile, depression is more commonly encountered in people with a chronic physical illness [18–21].

Participants who had been kept in solitary confinement experienced higher depression levels (significant association only on univariate level). Individuals in solitary confinement experienced isolation, sensory deprivation and inactivity [22]. Additionally, results indicated depression among participants who did not have Greek nationality. A possible explanation for this finding is that this group face language or cultural barriers, or worries about family and separation. Migrants experience psychological distress when prisons are not safe and when they do not have trusting relationships with fellow inmates [23].

According to the descriptive results, 38.6% of participants attempted suicide and 41.6% had harmed themselves. In North-Eastern Italy, data from 16 prisons with 3900 inmates during the period 2010–2016 showed suicide and attempted suicide rates of one and 15 per 1000 inmates, respectively. More than 90% of suicides and attempted suicides were of men between 21 and 49 years old, with most having committed violent crimes. Further, 14% of suicides and 19% of attempts had a previous history of suicide attempts and self-harm [24]. Inmates who attempted suicide reported higher levels of depression, hopelessness, impulsivity and aggression and lower levels of self-esteem and social support [25].

Zong et al. [26], exploring studies from 27 countries, including 35,351 suicides, demonstrated the suicidal ideation during the current period in prison, a history of attempted suicide and current psychiatric diagnosis as the strongest clinical factors associated with it. Institutional factors associated with suicide included occupation of a single cell and receiving no social visits. Criminological factors included remand status, serving a life sentence and being convicted of a violent offence, in particular homicide.

Suicidality may reflect an underlying vulnerability as defined by biological and psychological characteristics that increases under stressful situations in prison, which include general issues such as adaptation to prison, intense stress due to environmental factors, loss of liberty, withdrawal from a familiar environment as well as more specific aspects, such as violence, victimization, mistrust in an authoritarian environment, shame or guilt for wrongdoing, lack of control over the present and future, and absence of purposeful activity (work or education) [25,27]. Single-cell detention is the riskiest housing, with a suicide rate more than 400 times the rate of suicide in double-cell housing [28]. However, near-lethal attempts occur in early periods of custody, with hanging and ligaturing as the most prevalent methods used. Improved screening of suicide risk is required on arrival [29].

Moreover, descriptive results demonstrated sexual abuse in the case of 12.9% of participants and physical abuse in 54.5%, mainly by the father (49.1%). A relevant study in Canada showed a 21.9% prevalence of sexual abuse among male inmates [30]. The first episode of childhood sexual abuse among male inmates began at a mean age of $9.6 \pm 2.4$ years and ended at a mean age of $13 \pm 2.3$ years [31]. A relevant study in Greek showed experience of violence in parental relationships, mainly by the father and secondarily by the mother in 308 males in prison aged between 18 and 77 years [32]. Interestingly, incarcerated individuals often have a history of childhood abuse (physical and sexual) [30,33]. People abused in childhood are at a higher risk of perpetrating violence in adulthood and at higher risk of substance use, which could result in imprisonment. Unmet mental health needs associated with childhood abuse may exacerbate during imprisonment [30].

Furthermore, 35.6% of the present sample consumed alcohol prior to detention and 58.4% had a history of substance abuse. Growing up with a caregiver who used drugs or alcohol was a consistent predictor of increased risk of substance misuse [34]. Among male prisoners, drug abuse prevalence ranged from 10 to 48% and alcohol abuse disorders ranged from 18 to 30% [35]. Prisoners who committed homicide were likely to claim that it occurred under the influence of a substance. Meanwhile, one of the reasons people used substances was to increase their confidence in committing a crime [36]. Depression in male prisoners was strongly affected by the consumption of drugs in prison and by the absence of visitors. Consuming drugs while in prison was found in 41.4% ($n$ = 390) [37].

In the present study, 62.4% of the participants stated that someone was visiting them. The more a person feels connected to the social environment, the less likely they are to engage in self-destructive behaviors and suicide. Increasing social support may improve their ability to cope with stress that occurs during incarceration [37].

Finally, 50.5% of the sample studied reported future after prison as their major cause of stress, but this finding was not associated with depression. Considering life to be difficult after release from prison was associated with depression among 332 prisoners, of whom the majority were males ($n$ = 311, 93.7%) [38].

Apart from a psychiatric unit that is sufficiently equipped with materials and personnel, to detect and treat mentally ill inmates, equally important is the need to follow up in the community, after release. Mentally ill prisoners are faced with a double stigma after release, the one of imprisonment and that of a mental illness. After release, the unmet needs of this group may increase their risk of recidivism unless community psychiatric services are effective [10].

## 5. Strength and Limitations of the Study

The main limitation of this study is that it is a single-center study in Greece with a relatively small sample size which consists solely of male prisoners. The method of the present study was convenience sampling and therefore was not representative of all inmates in Greece. Additionally, it was a cross-sectional study, thus not allowing the emergence of a causal relation between depression and patients' self-reported characteristics. It would be interesting if there was a first evaluation of depression on admission and then periodically during the prison stay. Future research should take all of these limitations into consideration.

The strength of the study was that the Zung depression scale is a standardized and internationally recognized screening tool. This instrument may permit comparisons between inmates all over the world.

## 6. Conclusions

The results showed that foreign prisoners, those suffering from a chronic illness, or those kept in solitary confinement (only on an univariate level) were more depressed.

The results of the present study will contribute to a deeper understanding of male inmates' profile and suggest recommendations for interventions targeting the alleviation of depression. It is important to have an updated picture of mental health needs for the organization, planning, and delivery of services to this population as well as for informing policy and practice.

Inmates are a marginalized population who do not have the same benefits as individuals in the community, including health care. However, inmates ought to have access to available health services without discrimination, according to the universal basic principles of health. Prison staff training and the acquisition of behavioral skills to deal with the population is an urgent demand.

Future studies need to be conducted in a large sample size to assess the determinants of depression.

**Author Contributions:** Conceptualization, D.K. and M.P.; Methodology, D.K., E.D. and M.P.; Software, E.D. and N.P.; Validation, E.D. and A.K.; Formal analysis, E.D. and G.T.; Investigation, D.K., A.Z., N.P. and G.T.; Resources, N.P. and V.T.; Data curation, A.Z., A.K. and V.T.; Writing—original draft, D.K. and M.P.; Writing—review & editing, A.Z. and M.P.; Visualization, A.Z. and A.K.; Supervision, M.P. All authors have read and agreed to the published version of the manuscript.

**Funding:** This research received no external funding.

**Informed Consent Statement:** Informed consent was obtained from all subjects involved in the study.

**Conflicts of Interest:** The authors declare no conflict of interest.

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
