# Peer review of "Depression in Male Inmates"

_clinpract, doi:10.3390/clinpract13010001_

Round 1
Reviewer 1 Report
This study investigating depression among male inmates and the associated characters had clinical significance. However, the author's interpretation of the results is not rigorous. Especially, the discussion is not well written.
Introduction:
1. Introduction lacks a review of depression status in inmates.
2. “The prevalence of depression among prisoners in developing and developed countries is 39.2% and 33.1%, respectively.” Please cite references.
3. “Depression in men is associated with physical aggression, interpersonal conflict, risky activities, high alcohol and substance use, history of juvenile delinquency and being a repeat offender. “Please cite references.
Methods:
4. What’s the cut-off of Self-rating Depression Scale (SDS)-Zung? The authors mentioned several times that inmates had moderate depression. How was it detected?
5. There are a lot of questions about trauma in the interview, such as suicide and abuse. Are there any psychological services available if the participants feeling uncomfortable?
Results:
6. Table 2: Why were the years of imprisonment divided by below or above 10 years?
“Hours of stay in cell per day” divided by “more” or “less”, which was not rigorous.
7. Table 5
What's the difference between "yes" and "sometimes" in some questions (e.g., Fear of death)? The two options clearly overlapped.
8. The descriptions of depression were puzzling. What’s the cut-off of moderate depression? What percentage of people have low, or moderate, severe depressive symptoms?
9. Table 6 is not necessary.
Discussion:
10. Overall, discussion is not well written. The author's interpretation of the results is not rigorous. The structure of the discussion was also unclear.
11. “The present results illustrated moderate levels of depression among participants”. The results didn’t show the prevalence of moderate depression.
12. There are some superscripts in the first paragraph of the discussion (e.g., relevant study in Greece among male prisoners indicated that 223 (45.06%) 206 were diagnosed with a psychiatric disorder.4), but I can't find the corresponding notes.
13. There is great heterogeneity in the prevalence of depression among prisoners across studies. What causes it?
14. Many statements are unreasonable interpretations of the results. For example, “The statistical analysis showed depression among participants suffering from a chronic disease.”, “Male prisoners had moderate levels of depression”
15. “Participants who had been kept in solitary confinement experienced depression.” Table 8 shows that “Have you been in solitary confinement?” did not predict depression.
16. “Ethnicity may play a crucial role as individuals face with language barriers, worries about family and separation which often trigger hopelessness, depression, and anxiety.” Please cite references.
Conclusion:
17. “those kept in solitary confinement and those who had not thought about harming themselves were depressed”. Table 8 shows that “Have you been in solitary confinement?” did not predict depression.
Reviewer 2 Report
The researchers investigated depression and the associated characteristics among male inmates. Then researchers found the main factors associated with inmates’ depression and explored their impacts on depression. The design of this study is clear and the characteristics about inmates are comprehensive; the paper is well structured and clearly written. I have a few comments:
1. I was confused about the sentence “50% of inmates scored below 45 (median) on the SDS scale and 25% scored below 42 (median).” Was 42 the 25% quartile? Let me know if I misunderstand.
2. For the assessments of depression, how does the reliability of the scale?
3. The table 7 showed the association between depression and inmates’ characteristics. You reported some significant correlations and p-value, however, how do you get these results? For example, foreign inmates had higher levels of depression than Greeks, was it obtained by Mann-Whitney test?
4. In Discussion, authors state, “Participants who had been kept in solitary confinement experienced depression. Frequently, these individuals are more hard criminals or involved in incidents that indicate difficulties in adapting to prison.” In Results, solitary confinement was associated with depression but there was no statistically significant result from regression analysis. So, caution must be needed when interpret this result.
Reviewer 3 Report
The authors in this paper assess depression levels in male inmates in Greece. Overall, it is nicely written, although few typos.
The goal of the paper, however, is not clear to me. What is the message the authors want to deliver? Depression correlates? Depression incidence? What are the implications of your findings?
If there is a cut-off for depression in the lung scale (and I guess there is, see: https://www.sciencedirect.com/topics/medicine-and-dentistry/zung-self-rating-depression-scale), authors could estimate the prevalence of depression in their sample and compare it to the other samples reviewed in the discussion.
The paper needs more description of the statistical methodology and the authors should check whether their estimates are expressed in b or beta (see below).
######################
# Detailed comments
######################
Line 76:
how were assessed exclusion criteria? e.g., aggression
Line 80:
how was administered the survey? Was it a piece of paper or it was read by a doctor?
Line 85
There are several depression scales. Is there a reason for choosing specifically this one (Self-rating Depression Scale Zung (SDS))?
Line 92: visits by family and friends
Quantified in hours or frequency?
Line 109:
Is there a cut-off above which there is depression?
Table 7 and throughout:
Maybe Nationality would be better than Origin (Greek vs other)
Table 7:
It is not clear which type of regression is this one. Is it a model with y = depression and x1, x2 xn all the categorical predictors? Or is it a series of correlations (see line 168)?
This should be specified in the method section.
Line 179: Multiple linear regression was performed to assess...
How many? One or as many as your independent variables?
Line 181: Greeks (β = 2.77
If it's beta, the coefficient is expressed in SD and not in outcome units (i.e., depression). I suspect this is b and not beta (2.77 SDs is enormous and not reflected by CIs). Please check to be sure. Check all beta coefficient throughout the results
Line 183:
2.85 doesn't round to 2.8, but 2.9
Line 188 (whole paragraph):
It would be nice summarizing your findings.
Also, since the authors do a literature review on depression rates, it would be nice to provide the same figure for the current study. It is missing in fact a percentage of depressed inmates in your study.
Line 244: A research by Zhong et al.[25], in 77 studies from 27 countries
Maybe specify what kind or research (review, systematic review, meta-review)
Line 277 (whole paragraph):
Did the authors check whether in your sample there was a negative effect (i.e., depression reduction) of visits?
######################
# Minor comments
######################
Line 15:
maybe add "[The purpose ] of the study..." Otherwise it's a bit weird to read
Line 17: In the study were enrolled 101 male inmates
This is a (un)grammatical construction I see throughout the paper. Please check if it's correct. I'd suggest something more canonical, along these lines: "In the study, 101 male inmates were enrolled" (or a structure where subject verb and object are more evident)
Line 23: Statistically significant higher scores of depression had foreign prisoners compared to Greeks
It doesn't sound English to me, but maybe I'm wrong
Round 2
Reviewer 2 Report
The manuscript has been extensively improved. I don't have any other issues.